# Application of Machine Learning Techniques to Discern Optimal Rearing Conditions for Improved Black Soldier Fly Farming

**DOI:** 10.3390/insects14050479

**Published:** 2023-05-19

**Authors:** John Muinde, Chrysantus M. Tanga, John Olukuru, Clifford Odhiambo, Henri E. Z. Tonnang, Kennedy Senagi

**Affiliations:** 1International Centre of Insect Physiology and Ecology, Nairobi 30772-00100, Kenya; 2LabAfrica Research Centre, Strathmore University, Nairobi 59857-00200, Kenya; 3Sanergy Limited, Atlanta, GA 550288, USA

**Keywords:** insect for feed, insects farming, black soldier fly, Internet of Things, machine learning

## Abstract

**Simple Summary:**

In recent years, farming the black soldier fly (BSF) *Hermetia illucens* (L.) (Diptera: Stratiomydiae) has gained popularity across the globe due to its usefulness mainly in animal feed production and waste management. The short cycle time taken to rear the BSF and the high protein content present in its larvae makes it a suitable source of feed for a variety of animals (e.g., poultry, fish, and pigs); the livestock bred as food for humans. However, despite the farming of black soldier fly larvae (BSFL) as a source of feed, its production levels are low and do not meet the existing market demand. This study explored data science and machine learning modeling approaches to discern optimal rearing conditions for improved BSFL farming. The random forest regressor machine learning algorithm gave the best prediction results. The algorithm also ranked the variables that contributed most to the prediction of the expected larvae weight. Given the studied rearing conditions, the prediction algorithm can discern and predict the expected weight of BSFL to be harvested. Tuning the production system parameters according to the order of the ranked parameters can further optimize the production of BSFL. BSFL are a source of feed for the animals that are a source of food for humans; therefore, this research contributes to alleviating food insecurity.

**Abstract:**

As the world population continues to grow, there is a need to come up with alternative sources of feed and food to combat the existing challenge of food insecurity across the globe. The use of insects, particularly the black soldier fly (BSF) *Hermetia illucens* (L.) (Diptera: Stratiomydiae), as a source of feed stands out due to its sustainability and reliability. Black soldier fly larvae (BSFL) have the ability to convert organic substrates to high-quality biomass rich in protein for animal feed. They can also produce biodiesel and bioplastic and have high biotechnological and medical potential. However, current BSFL production is low to meet the industry’s needs. This study used machine learning modeling approaches to discern optimal rearing conditions for improved BSF farming. The input variables studied include the cycle time in each rearing phase (i.e., the rearing period in each phase), feed formulation type, length of the beds (i.e, rearing platforms) at each phase, amount of young larvae added in the first phase, purity score (i.e, percentage of BSFL after separating from the substrate), feed depth, and the feeding rate. The output/target variable was the mass of wet larvae harvested (kg per meter) at the end of the rearing cycle. This data was trained on supervised machine learning algorithms. From the trained models, the random forest regressor presented the best root mean squared error (RMSE) of 2.91 and an R-squared value of 80.9%, implying that the model can be used to effectively monitor and predict the expected weight of BSFL to be harvested at the end of the rearing process. The results established that the top five ranked important features that inform optimal production are the length of the beds, feed formulation used, the average number of young larvae loaded in each bed, feed depth, and cycle time. Therefore, in that priority, it is expected that tuning the mentioned parameters to fall within the required levels would result in an increased mass of BSFL harvest. These data science and machine learning techniques can be adopted to understand rearing conditions and optimize the production/farming of BSF as a source of feed for animals e.g., fish, pigs, poultry, etc. A high production of these animals guarantees more food for humans, thus reducing food insecurity.

## 1. Introduction

It is estimated that the world population will grow to approximately 8.5 billion by 2030 and 9.7 billion by 2050; on the African continent, it is expected that 26 out of the 54 countries in the region will experience a population increase that is at least double their current size by 2050 [1]. Therefore, there is a need to come up with an alternative, sustainable and innovative approach to provide food to meet the growing demand for food due to the increasing human population [2]. One possible solution to address this challenge is through insect farming, in particular, the black soldier fly (BSF) *Hermetia illucens* (L.) (Diptera: Stratiomydiae) larvae as a source of feed for farmed animals which in turn are used to provide food for humans [3,4,5,6,7,8].

Black soldier fly larvae (BSFL) stand out from other commonly farmed insects due to several reasons. They have a short maturity period (i.e., it takes 10–15 days for the eggs to turn into mature larvae) [9]. This implies that farmers can obtain a return on investment within a short period of time. The dry or wet BSFL can be harvested and fed to animals without further processing [10]. In addition, BSFL is considered not to be a pest or vector of any diseases and, therefore, does not cause any significant nuisance [11]. BSFL grease is a suitable raw material to produce biodiesel [12]. BSFL have bioactive compounds, such as chitin and antimicrobial peptides, which have high biotechnological and medical potential [13]. BSFL have also been used for bioplastic production [14]. BSFL are also rich in protein, fat and amino acids, and are hardly affected by diseases. Their ability to feed and grow in a wide variety of organic residues in addition to providing a sustainable and reliable source of feed for animals to feed humans is a key step towards addressing the challenges of food insecurity [6]. However, the current market production of BSFL for feed is quite low compared to the existing demand [4,5,13,14]. This underscores the need to investigate other approaches that can contribute to improving the rearing process of BSFL to improve its production as a source of feed for animals. Moreover, farming BSFL addresses other challenges, such as fertilizer production [10] and waste management [15,16].

As a result, this research employed a machine-learning-driven approach to investigate the rearing parameters with the aim of incorporating data-driven decision making in the farming of BSFL to optimize its production. In this context, the term optimize refers to the best possible combination of rearing parameters that result in maximum weight harvest of the larvae based on the rearing process. The BSFL farming process at the (anonymized) factory is mechanized and encompasses the use of Internet of Things (IoTs) devices to collect data and monitor larval growth. This study utilized the data collected by IoTs and employed data science and machine learning approaches to generate insights for the purpose of facilitating data-driven decision making in addition to coming up with a model to optimize and predict the harvested larvae weight.

## 2. Materials and Methods

### 2.1. Data Collection Procedure

Data were collected from a mechanized BSF factory (anonymized), from the month of February to September 2022, with the assistance of domain experts. In the factory setup, the rearing process was divided into two phases (1 and 2). Thereafter, the BSFL were harvested. The total mass was measured using a weighing balance, and the mass was recorded separately for each bed. The final dataset had 14 variables; 13 input variables and 1 target variable. In this context, the target variable refers to the mass of larvae harvested, while input variables refer to the features that impact the mass of larvae harvested. The variables are explained in Table 1 below.

In Phase 1, the hatched eggs were reared for a period of 3–7 days on different substrate (i.e., Formulation C and SW) compositions. After that, the larvae were then moved to Phase 2, where they were introduced to a different substrate (Formulation SM and B) composition. The larvae stayed in Phase 2 for 7–10 days. Just before the larvae transitioned into the pre-pupae stage, the BSFL per bed were harvested by sieving them from the substrate, and washed. Their wet weight was measured using a weighing balance and recorded. The weighing balance used in measuring the weight was a highly customized Yaohua A12 electronic weighing scale (model number TCS2-A12E).

### 2.2. Data Processing Techniques

This process entails cleaning the dataset for purposes of analysis. Different procedures can be used in this process, including handling missing values (empty entries in the dataset, and such entries are also known as null values), outliers (data entries that are either too small or too large compared to other values in the dataset), and duplicates (repeated entries). Section 2.2.1 discusses the techniques used in handling missing values. Outlier treatment approaches are explained in Section 2.2.2, while feature transformation approaches are covered under Section 2.3.1.

#### 2.2.1. Handling Missing Values

Missing data values refers to random null values (empty entries) within the dataset. In this context, this could have been due to technical errors during the data collection process. The following items discuss the specific missing data problems that were identified in the dataset and how they were imputed/calculated:(a)Missing completely at random (MCAR): MCAR happens when missing entries are randomly distributed across variables and are unrelated to other features. Kenyhercz and Passalacqua [17] came across the same challenge in their dataset. To solve this problem, they compared 4 different imputation techniques that included hot deck, iterative robust model, k-nearest neighbor (KNN), and variable means. Their results showed that KNN was the most accurate approach, as it had the lowest error between the imputed and actual values. Actual values refer to complete data entries collected during the data collection process, while imputed values refer to entries calculated to fill up missing values in the collected dataset. Technically, the cluster-based approach, using an appropriate algorithm such as KNN, entails identifying the nearest neighbors of the record with a missing entry, imputing the value of the nearest neighbors, and using the result to fill up the null record. It is for this reason that this research adopted this approach in imputing missing values. The actual variables evaluated using this approach were Phase 1 feed depth and total young larvae (YL) mass loaded (kg). This was implemented using sklearn’s KNNImputer [18].(b)Missing at random (MAR): This study also identified missing at random (MAR) data entries. In MAR, the missing data in one variable can be explained by another variable in the same dataset. For this study, the variables imputed using this approach include the feeding rate and the number of young larvae loaded at the first phase. To solve this, this research integrated the weight class adjustment approach. It was implemented using the group-by methodology [19] based on related records from other variables in the dataset to fill up the missing entries. This approach was also used by other researchers, such as [19], to solve similar problems.

#### 2.2.2. Outlier Treatment

An outlier is an extreme value that is isolated from other values. In this study, outliers were not removed from the dataset; instead, a flooring and capping approach was used to replace the outliers with the respective floor and capping computed values. Flooring entails replacing all the values that fall below the selected minimum in a data column with a calculated value that falls within the set range, while capping entails replacing all the higher-side values that fall above the selected maximum [20]. Table 2 shows how this research checked outliers against each input variable and how they were handled. The techniques employed were flooring and capping, which have been used by other researchers such as [21] to handle outlier entries. These techniques involve limiting the extreme variables to fall within the selected percentiles of the maximum and minimum variables [20]. The boxplot shown in Figure 1 and Figure 2 shows an illustration of how outliers were detected and resolved, respectively, on the actual Phase 1 feed depth variable, as an example, using the flooring and capping approaches. Values below the lower percentile (*Lp*) and values above the upper percentile (*Up*) were considered to be outliers and were floored and capped respectively. The ranges for flooring and capping percentiles, shown in Table 3, were selected depending on the outliers identified on each variable after visualization on a boxplot. The values presented in Table 3 are important to the study, as they outline the lower and upper capping percentiles for the transformed variables. The flooring and capping formulae are shown in Equations (Equation 1) and (Equation 2), respectively, where *df* represents the data frame, *x* is the variable of interest, and $y_{0}$ and $y_{1}$ are the percentiles. The flooring and capping techniques were used to adjust the values since they do not result in the deletion of records, to the advantage of this research.
(1)$$Up=df{[}^{\prime }x^{\prime }].quantile\left(y_{0}\right)$$
(2)$$Lp=df{[}^{\prime }x^{\prime }].quantile\left(y_{1}\right)$$

### 2.3. Machine Learning Modeling

#### 2.3.1. Feature Transformation

Feature transformation involves techniques that convert variables to be useful for further analysis. For instance, one-hot encoding was integrated to convert categorical variables (such as Phase 1 and 2 formulation, and Phase 1 purity score) into numerical data for machine learning. One-hot encoding is a technique that is used to convert categorical data to numerical data to facilitate the learning process since machine learning models are incapable of processing plain texts in their raw form [22]. In this study, one-hot encoding was implemented using get_dummies() in the sklearn Python library.

Feature splitting was used in cases where variables contained information that could be split into two or more columns. In the context of this study, feature splitting refers to the separation of a variable into two distinct columns. For example, the date variable comprising the month and year of the rearing process was split into years and months to create two new distinct columns of months and years, thus facilitating a comparison of production performance within different month periods.

This research notes that the bed length and width were measured in meters (m). At the end of the rearing process, the weight of the BSFL harvested was measured in kilograms (kg). In this context, feature aggregation, a technique used to combine features with the aim of creating a robust feature, was adopted. For instance, the weight in kilograms (represented as ω), and the product of bed length and width (represented as φ) were used to calculate the target variable (represented as *y* and expressed in kg/m${}^{2}$) using Equation (Equation 3):(3)$$y=\frac{\omega }{\varphi }$$

#### 2.3.2. Data Splitting

The data was split into a ratio of 70:20:10. In total, 70% of the data was used to train the models, 20% was used for validation and 10% was used as the test set. Figure 3 shows the data-splitting and model evaluation processes.

#### 2.3.3. Feature Scaling

Feature scaling is a data pre-processing technique that transforms numeric features of the dataset to be within a specified range with the aim of improving the performance of machine learning models [23]. In this study, feature scaling was carried out on numerical variables (in Table 2), and values were set to fall within a range of −1 and 1, using Z-score normalization, expressed in Equation (Equation 4):(4)$$z=\frac{x-u}{\sigma }$$

#### 2.3.4. Trained Machine Learning Models

After completing data prepossessing and preparation, the next step entailed training/fitting approximately seven supervised machine learning models that are inbuilt and available in the Scikit-learn Python library [24].

The trained machine learning models were linear regression, ridge regression, decision tree regressor, random forest regressor, bagging regressor, adaboost regressor, and XGBoost regressor. All the models were trained with the understanding that the best model would be selected. Sklearn’s Gridsearchcv was used to select the best combination of hyperparameters for performance tuning. From these, the best-performing machine learning model was selected and compared with those from existing literature. Previous studies that applied machine learning algorithms to investigate various BSFL growth phenomena include generalized linear model [25] and statistical modeling [26].

#### 2.3.5. Performance Evaluation Criteria

To evaluate the developed models, five metrics were selected to measure the performance of machine learning algorithms. The metrics were, namely, the root mean squared error (RMSE), the mean absolute error (MAE), the mean absolute percentage error (MAPE), R-squared, and adjusted R-squared, which are defined in Equations (Equation 5)–(Equation 9), respectively. In the equations, *N* is the number of observations in the dataset, $\hat{y_{i}}$ is the predicted values, $y_{i}$ is the actual values, and *p* is the number of predictors (independent variables).

RMSE, defined in Equation (Equation 5), is expressed as the standard deviation of the residuals (errors), which are the differences between the predicted and actual values. As such, it is calculated by obtaining the square root of the average of the squared residuals [27]. A lower RMSE implies better model performance, while higher RMSE implies poor model performance [28]:(5)$$RMSE=\sqrt{\frac{1}{N}\sum_{i=1}^{N}{\left(\hat{y_{i}}-y_{i}\right)}^{2}}$$

MAE, defined in Equation (Equation 6), calculates the absolute values of each prediction error [27]. A lower MAE implies good performance, while a higher MAE implies poor performance [28]:(6)$$MAE=\frac{1}{N}\sum_{i=1}^{N}\left|y_{i}-\hat{y_{i}}\right|$$

MAPE, defined in Equation (Equation 7), measures the absolute percentage difference between predicted and actual values. It is calculated by first obtaining the difference between actual values and the predicted values divided by the actual values, followed by obtaining the mean of the results [29]. A lower MAPE means good model performance, while a higher MAPE implies poor performance [28]:(7)$$MAPE=\frac{1}{N}\sum_{i=1}^{N}\left|\frac{\left(y_{i}-\hat{y_{i}}\right)}{y_{i}}\right|$$

R-squared, defined in Equation (Equation 8), evaluates the proportion of variance in the predicted (dependent) variable that can be explained by the predictor (independent) variables. It essentially represents the goodness of fit of the model [30]. Adjusted R-squared, defined in Equation (Equation 8), is a modified version of R-squared. The main difference is that adjusted R-squared accounts for predictors that are not significant when evaluating the goodness of fit, while R-squared does not account for predictors that are not significant. In the context of this study, predictors are the variables described in Table 1, excluding the mass harvested variable, which is the target to be predicted. As such, adjusted R-squared evaluates whether adding additional predictors results in the improvement of the model or not [30]. For R-squared and adjusted R-squared, higher scores imply good model performance, while lower scores imply poor model performance [28]:(8)$$R^{2}=1-\left[\frac{\sum_{i=1}^{N}\left(y_{i}-\hat{y_{i}}\right)}{\sum_{i=1}^{N}\left(y_{i}-\bar{y_{i}}\right)}\right]$$
(9)$$R_{adj}^{2}=1-\left[\left(1-R^{2}\right)\frac{N-1}{N-p-1}\right]$$

Although each of the above metrics gives insights into the performance of the models, a complete assessment of the model performance requires a combination of more than one metric. As such, the model performance in this study was assessed by generating scores for each of the above metrics and picking the highest-performing model based on the best combination of the scores. Essentially, when comparing models, the lower values of RMSE, MAE, and MAPE are desired, while the higher values of R-squared and adjusted R-squared are better.

#### 2.3.6. Ranking of Features

To rank the features in terms of their importance (i.e., key features that inform the target variable) for the prediction of BSFL expected weight, this study utilized the inbuilt feature importance function in the Scikit-learn [24] Python library for specific algorithms. For example, Equation (Equation 10) calculates the important features in the random forest regressor algorithm. In the equation, *f*, *t*, and *m* represent the features, trees in the random forest regressor, and nodes, respectively. The Gain function is the inbuilt function used to compute the feature importance [24]:(10)$$Importance_{f}^{t}=\frac{\sum_{m\in M_{f}^{t}}Gain_{m}}{\sum_{f}\sum_{m\in M_{f}^{t}}Gain_{m}}$$

## 3. Results

### 3.1. Data Exploration

Figure 4 shows a correlation heat map of the mass harvested, Phase 1 bed length, and Phase 1 feed depth. It is observed that there exists a negative (r=−0.6) correlation between the bed lengths and the mass harvested.

In addition, this research compared all the input variables (outlined in Table 1) that contributed to the minimum and maximum masses of larvae harvested, represented in Figure 5. As evident in Figure 5, the maximum mass of larvae harvested in terms of weight was 48.381 kg/m${}^{2}$, while the minimum mass of larvae weight harvested was 5.793 kg/m${}^{2}$, highlighting a difference of approximately 42.588 kg/m${}^{2}$.

### 3.2. Performance Evaluation of Machine Learning Models and Ranking of Variables

The best model was selected based on the comparison of the five metric scores described in Section 2.3.4. Table 4 and Figure 6 give the metrics comparison scores for each of the models. It is seen that the random forest regressor performed better than the other models with RMSE, MAE, R-squared, adjusted R-squared, and MAPE values of 2.912162, 2.221066, 0.809009, 0.791275, and 14.661590, respectively. The performance of the random forest regressor was compared to that of algorithms deployed in previous works (i.e., statistical model (ordinary least square), and generalized linear model), and the results are tabulated in Table 5. It was also seen that the random forest regressor produced the best results.

As shown in Figure 7, the top five important features identified by the best-performing model (random forest regressor) were Phase 1 and 2 bed lengths, Phase 1 Formulation C, the average number of larvae in Phase 1 per gram, and total YL mass loaded.

## 4. Discussion

### 4.1. Data Analytics

In Figure 4, the mass harvested increases with a decrease in the bed length. This means that a decrease in the Phase 1 bed length (probably to approximately 16 m, as shown in Figure 5) gives an optimal production (probably to 48.381 kg/m${}^{2}$, as shown in Figure 5). On the other hand, there was no (r=0.0) correlation between the feed depth and the mass harvested as shown in Figure 4. This means an increase/decrease in the depth of the beds does not result in the increase/decrease in harvested larvae and therefore is not an important parameter to tune to optimize production.

In Figure 5, there is a slight difference between the total young larvae loaded at the start of the rearing process (0.85 kg at for the maximum mass harvested and 0.83 kg for the minimum mass harvested). This is an important metric to monitor in order to limit competition for resources in the rearing beds. In addition, the bed length in Phase 2 was shorter (10.5 m) for the maximum weight, while for the minimum weight, it was long (46.0 m), pointing to shorter beds being more efficient in terms of expected output.

Moreover, as is evident in Figure 5, it was observed that the cycle time in Phase 2 was shorter (3 days) for the maximum weight, while for the minimum weight observed, the cycle time was longer (6 days). This is in line with other studies [31] that observed that the expected harvest weight of the larvae decreases as the cycle time increases. This is because, as observed in previous studies [32], the larvae tend to stop feeding as they approach the prepupae stage, and this results in reduced weight gain since they start utilizing internal food reservoirs. In addition, when there was maximum weight, it was noted that the larvae were not fed sour worms (coded as SW) but instead were fed substrates rich in protein and carbohydrates (coded as Formulation C). It was also observed that larvae feed substrates rich in carbohydrates and low protein ratio (Phase 1 Formulation C and Phase 2 Formulation SM) had high overall weight gain compared to larvae fed on substrates that had low protein and carbohydrate content. This is in line with studies carried out [33,34] to investigate the impact of the substrate content on BSFL performance.

In terms of the feeding rate, it can be seen from Figure 5 that for the maximum mass harvested, the larvae had a higher feeding rate of approximately 63 milligrams per young larvae per day compared to 45 milligrams per young larvae per day in the minimum mass harvested. This informs us that this could be the optimal feeding rate for optimal production.

### 4.2. Machine Learning Algorithms and Feature Ranking

The random forest regressor performed better compared to all other algorithms in terms of the RMSE, MAE, R-squared, and adjusted R-squared scores. However, besides XGBoost regressor having a better score on MAPE only compared to the random forest regressor, it performed relatively poorly on all other metrics. Therefore, this study selected the random forest regressor as the best-performing algorithm, with MAPE, MAE, RMSE, R-squared, and adjusted R-squared values of 14.66%, 2.22, 2.911, 80.9%, and 79.12%, respectively. Therefore, this study is confident that the random forest regressor has the ability to discern and predict the expected weight of the BSFL to be harvested. The random forest regressor results were compared to models deployed in previous works and tabulated in Table 5. In this context, the random forest regressor gave the best results of RMSE, MAE, $R^{2}$, and adjusted $R^{2}$; hence, picked as the best machine learning model for predicting the expected harvest of BSFL weight.

As shown in Figure 7, the random forest regressor model informs us that these were the top features that most influence the computation and prediction of the mass of BSFL harvested. Therefore, as a decision support system, the algorithms can advise the farmer to tune the input variables in that order of importance in order to achieve optimal production. The optimal levels to tune the variables were discerned and discussed in Section 4.1. Out of the top five important features, it is observed that Phase 2 bed length was ranked with the highest feature importance score in terms of predicting expected harvest compared to the other variables. Indeed, experts mentioned that monitoring and managing shorter beds was easy compared to the long ones and led to higher production.

## 5. Conclusions

This study employed data science and machine learning approaches to investigate the optimal rearing parameters to monitor and improve black soldier fly larvae farming. These approaches discerned the values (e.g., the loaded length of the beds for Phase 1 and Phase 2 to be 45 m and 35 m, respectively) of input variables that gave optimal production. Therefore, a farmer should tune the input variables within those levels to achieve an optimal harvest. Moreover, this research trained seven machine learning models to predict the expected weight of BSFL to be harvested. The random forest regressor was found to be the best-performing model. Consequently, during the rearing process, the farmer can measure the parameters and obtain the projected/predicted expected BSFL weight to be harvested, i.e., a decision support system. As such, a farmer can tune the production system input variables to the expected levels to obtain an optimal harvest at the end of the rearing process. Moreover, the random forest regressor ranked the input variables studied in order of importance. Among others, the length of the bed was found to be the most important parameter to tune, meaning that it was necessary to tune the input variables as per that order of ranking since it contributed the most to high production. In the future, this study proposes to expand the analysis by combining the current rearing variables with environmental variables. This will facilitate a detailed investigation of how the combined features affect the growth and production of black soldier fly larvae to improve/optimize the farming process. In conclusion, these approaches can be widely adopted in the farming of black soldier fly larvae as a source of feed for animals (e.g., pigs, fish, and poultry), which in turn are a source of food for humans, hence contributing to reducing food insecurity.

## Figures and Tables

**Figure 1 insects-14-00479-f001:**
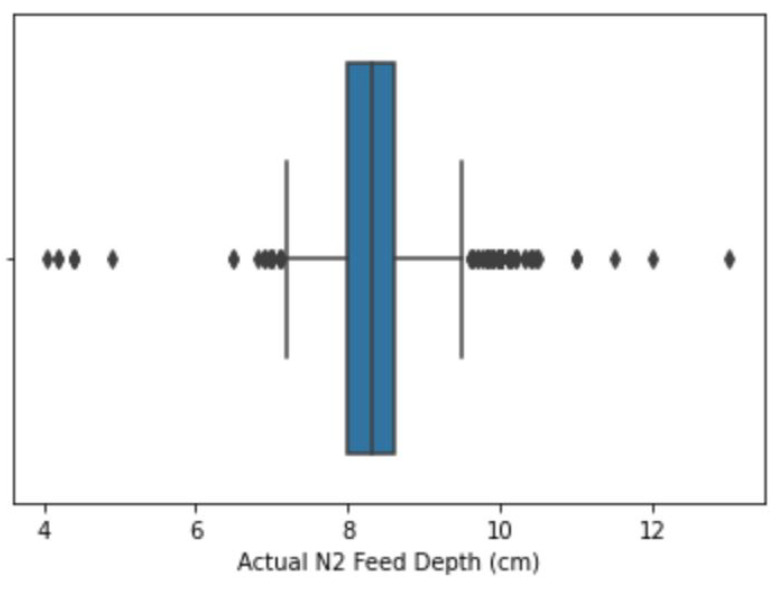
Outliers present.

**Figure 2 insects-14-00479-f002:**
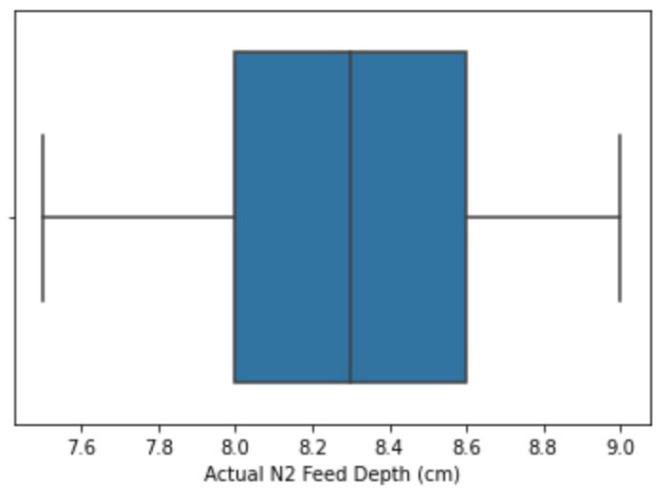
Outliers resolved.

**Figure 3 insects-14-00479-f003:**
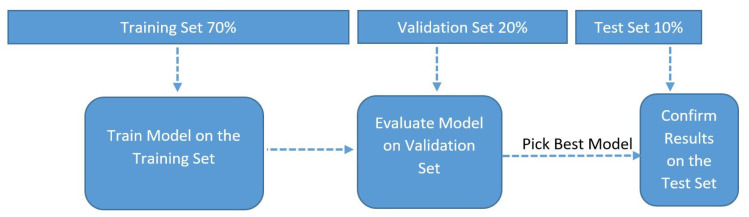
Data-splitting and validation approaches.

**Figure 4 insects-14-00479-f004:**
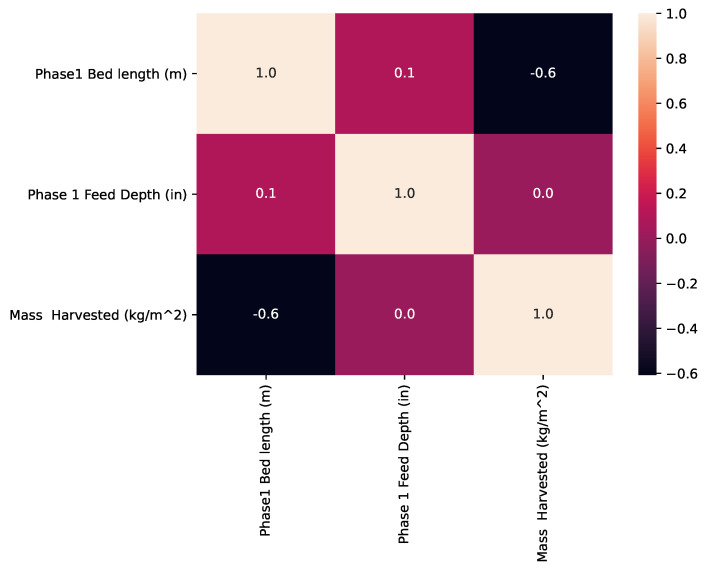
Correlation analysis of the bed length, and feed depth against mass harvested.

**Figure 5 insects-14-00479-f005:**
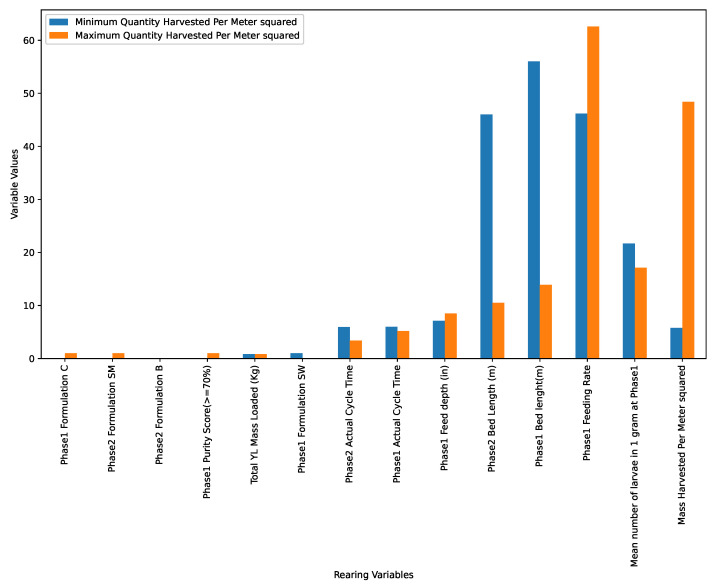
All the input variables that contributed to the minimum and maximum masses of BSFL harvested.

**Figure 6 insects-14-00479-f006:**
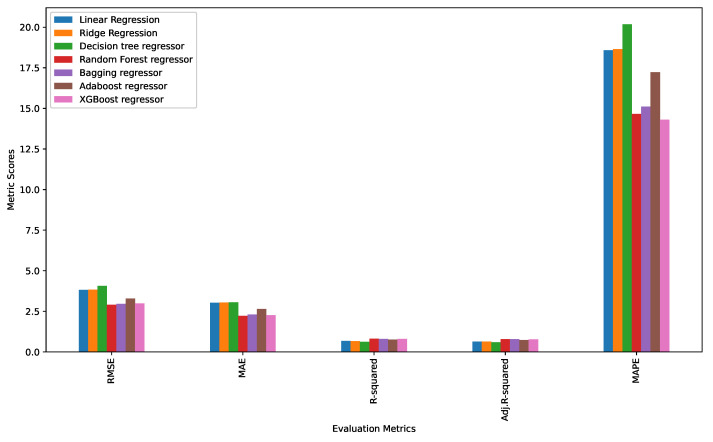
Graphical illustration of the performance metrics of the trained machine learning algorithms.

**Figure 7 insects-14-00479-f007:**
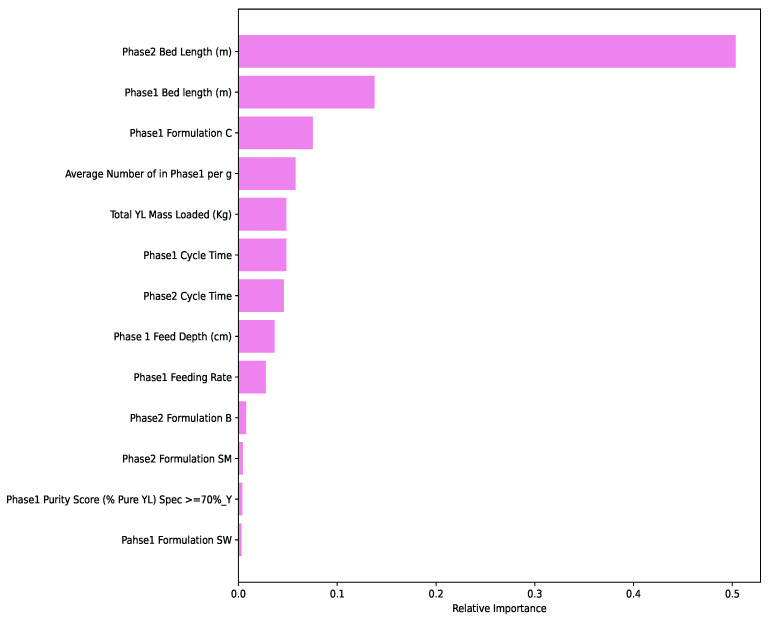
Important parameters that affect the rearing of BSFL.

**Table 1 insects-14-00479-t001:** A description of input variables and the target variable. Note that the mass harvested is the target variable, while the others are input variables.

No.	Variable Name	Description
1	Phase 1 Cycle Time (days)	Time taken to rear the larvae in the first phase.
2	Phase 1 Formulation C	Coded name of feed type used in Phase 1.
3	Phase 1 Bed Length	length of the rearing area at Phase1.
4	Phase 1 Purity Score	percentage purity recorded after separating the larvae from the substrate.
5	Phase 1 Formulation SW	Coded name of a feed used in Phase 1.
6	Average Number of Larvae in Phase 1 in 1 g	Estimated number of young larvae per 1 g.
7	Phase 1 Feed Depth (in)	Measurement of the feed depth.
8	Phase 1 Feeding Rate in milligrams (mg) per young larvae (YL) in a day, i.e., mg/YL-Day.	This refers to the estimated mass of feed given to each larva per day. The young larvae refer to those that were 2–3 days old.
9	Phase 2 Formulation SM	Coded name of feed type used in Phase 2.
10	Phase 2 Bed Length (m)	Length of the beds at the second phase.
11	Phase 2 Formulation B	Coded name of the feed type used.
12	Phase 2 Cycle Time (days)	Rearing time under Phase 2.
13	Total Young Larvae (YL) Mass Loaded (kg)	Amount of larvae added in mass at each bed.
14	Mass Harvested	Refers to the amount of larvae harvested in kilograms (kg) at the end of the rearing cycle comprising of phase 1 and phase 2.

**Table 2 insects-14-00479-t002:** Outlier checks and appropriate treatment.

No.	Variable Name	Outlier Treatment
1	Phase 1 Actual Cycle Time (days)	Outliers were corrected using the flooring and capping approach. Percentile values are shown in Table 3.
2	Phase 1 Formulation Coke	Categorical variable; no outlier score.
3	Phase 1 Bed Length	Outliers were corrected using the flooring and capping approach. Percentile values are shown in Table 3.
4	Total Young Larvae (YL) Mass Loaded (kg)	No presence of outliers.
5	Phase 1 Purity Score	Boolean (Yes/No) variable; no outlier score.
6	Phase 1 Formulation SW	Categorical variable; no outlier score.
7	Average Number of Larvae Phase 1 in 1 g	Outliers were corrected using the flooring and capping approach. Percentile values are shown in Table 3.
8	Phase 1 Feed Depth (in)	Outliers were corrected using the flooring and capping approach. Percentile values are shown in Table 3.
9	Actual Phase 1 Feeding rate (mg/YL-Day)	Anomaly outliers not detected.
10	Phase 2 Bed Length (m)	No presence of outliers.
11	Phase 2 Formulation B	Categorical variable; no outlier score.
12	Phase 2 Formulation SM	Categorical variable; no outlier score.
13	Phase 2 Cycle Time (days)	Outliers were corrected using the flooring and capping approach. Percentile values are shown in Table 3.
14	Mass Harvested	This variable was not treated for outliers to avoid altering the distribution of the target variable.

**Table 3 insects-14-00479-t003:** Outlier flooring and capping percentiles.

Variable Name	Lower Percentile	Upper Percentile
Phase 1 Cycle Time (days)	0.01	0.99
Phase 1 Bed Length	0.05	0.95
Phase 1 Average Number of Larvae in 1 g	0.02	0.98
Phase 2 Cycle Time (days)	0.01	0.99

**Table 4 insects-14-00479-t004:** Metrics scores for the trained machine learning algorithms.

	RMSE	MAE	R${}^{2}$	Adjusted R${}^{2}$	MAPE
Linear Regression	3.818960	3.027252	0.671549	0.638467	18.587188
Ridge regression	3.832344	3.043502	0.669242	0.638529	18.645440
Decision tree regressor	4.073082	3.048241	0.626382	0.591689	20.185357
Random forest regressor	2.912162	2.221066	0.809009	0.791275	14.661590
Bagging regressor	2.961843	2.298820	0.802437	0.784092	15.113145
Adaboost regressor	3.29355	2.647179	0.755707	0.733023	17.228926
XGBoost regressor	2.990458	2.255420	0.798601	0.779900	14.305270

**Table 5 insects-14-00479-t005:** Metrics scores of the reviewed machine learning algorithms.

	RMSE	MAE	R${}^{2}$	Adjusted R${}^{2}$	MAPE
Statistical model (ordinary least square)	3.818960	3.027252	0.671549	0.638467	18.587188
Generalized linear model	3.855807	3.087771	0.665180	0.634089	18.712122
Random forest regressor	2.912162	2.221066	0.809009	0.791275	14.661590

## Data Availability

The data used in the current study are available upon request by contacting the International Centre of Insect Physiology and Ecology. The codes for processing the data are available here [35].

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
