# Peer review of "Application of Machine Learning Techniques to Discern Optimal Rearing Conditions for Improved Black Soldier Fly Farming"

_insects, 2023, doi:10.3390/insects14050479_

Round 1

Reviewer 1 Report (Previous Reviewer 1)

Clearly, this is a better version of the manuscript than the previous one.  But there are still several issues requiring the attention of the authors prior to the publication of this work.

My biggest concern with this manuscript relates to the lack of clarity of many details in the methods and results sections, and on the interpretations of results.  It is vital that readers can clearly understand how the work was performed and how the models were fitted to the data and how the overall modeling work was conducted.  There is a severe lack of statistical details that prevent a full understanding of the methods followed for model fitting.  Or there are mixed sections describing methods and results together.  Having many gaps in the understanding of the methods limits understanding and appreciation of the results.  Speaking of the results, I think authors can do better in describing the patterns they found.  For instance, in lines 170-176, the description and interpretation of results is confusing because Figure 4 does not show the relationship between harvest and bed length, or between harvest and feed depths.  It is required to perform additional statistical tests (e.g., correlation analyses) to support these statements of a relationship between variables.  The description of the results should be limited to what authors actually show in their figures.

The manuscript reports that the best-performing model was the random forest regressor, in terms of MAPE, RSM, MAE, R-squared and Adjusted R-Squared statistics (lines 206-209).  However, according to Figure 7, there was no clear distinction in the score values of the r, R^2 and RMSRE statistics calculated for the different models.  In the case of the RMSE statistic, the Ridge Regression and the decision tree regressor model separate the most from the other models.  Just in the case of the MAPE statistic, there is a clear separation among the fitted models with the lowest value estimated for the XGBoost regressor model.  Therefore, if lower MAPE values are better, then the XGBoost regressor model should be deemed as good (or even better) than the Random forest regressor model proposed by the authors.  Should this be the case, additional work will be required in the analyses, results, discussion and conclusions.

It is mentioned that the study aimed to discern optimal rearing conditions, but it is unclear how, exactly, the study suggests reaching "optimality".  In the conclusion, I was expecting to find a stronger closing statement relating to the optimal conditions that the authors would propose to the anonymized factory to improve their production, based on the machine learning modeling work.  But this is not included in the manuscript.  Please indicate what are the optimal conditions the authors propose for the anonymized factory.  Give hints on how the production of flies should be improved following your suggestions.  After all, the manuscript from the title indicates that it deals with optimization, and not finding a conclusion regarding optimization of the conditions for rearing, leaves a feeling of emptiness when you finish reading the manuscript.

In my opinion, there is a need for this manuscript to improve the description of the statistical methods, results and discussion sections, because otherwise, they will limit readers’ appreciation of the research findings and of the value of the study.  In my opinion, there are various issues that limit the publication of the manuscript in its current form.

Below I provide additional comments that I think could be helpful for authors to further improve their manuscript.

L 4, suggest deleting "In addition", and starting the sentence as "The Black Soldier Fly larvae..."

L 8, "explored" or instead "used"?

L 9-11, "cycle time in each rearing phase", "purity score in the first phase" and "length of the beds at each phase", is there another way authors can explain these variables more simply and easily to understand?  The way it is presented is ambiguous and will mean nothing to many (if not most) readers.

L 12, suggest "harvested (kg per meter)" instead of "harvested in kg per meter", and indicate "per meter" of what.

L 17, "loaded length of the beds", could you explain this simpler?

L 29, after "... human population" include citations to support this statement.  In a previous review, I asked authors to cite relevant literature where appropriate, but it seems that they insist on presenting information without recognition of relevant literature on the topic.

L 38-39, include citations in "BSFL grease is a suitable raw material to produce biodiesel".

L 41-45 Include citation.

Text in lines 61-63, is not necessary and can be safely removed.

L 64 add "Procedure" (i.e., Data Collection Procedure).

In section 2.1, it would be nice if the authors provide information about the BSF production at the anonymized factory.  For instance, "The BSF anonymized factory produces __ million larvae per week for which __ tons of larval diet are used”, or something like that that allows an understanding of the magnitude of production of the anonymized factory.

Table 1 needs editions to improve clarity.  It is not clear what “variable No. 4” reflects and how it can be interpreted.  It could help to improve readers’ understanding if you explain exactly what "the set parameters" are.  In variable No. 8, the units of the variable name appear like "(mg/YL-Day)", but first you need to indicate what "YL" means (I guess it is young larvae, but it is not the responsibility of readers to be guessing what authors want to communicate); it is also confusing that such units seem to indicate that this variable was measured as the mass (mg) that each individual larvae feed during a certain period, but the description of the variable is expressed as the number of times the larvae are feed, which is a different concept.  You need to clarify this.

L 71-72, indicate the exact substrates used (C, SW, other).

L 73, indicate the exact feed composition used (B, SM, other).

L 74-76, indicate how the harvesting and weighing of larvae were performed.

L 76 add "Techniques" (i.e., Data Processing Techniques).

L 78, describe null values, outliers, and duplicates the first time you mention these terms in the manuscript.  For example: "null values (i.e., __), outliers (i.e., __), and duplicates (i.e., __)".

L 78-79, "Section 2.2.1 discusses the techniques used in imputing null values...", but Section 2.2.1 is labeled as "2.2.1. Missing Values".  You need to define and describe terms and be consistent with their use throughout the manuscript.

L 83, it is not clear what the authors refer to with "null values".  Again, you need to define and explain terms so that readers can understand what you are referring to.  I insist it is not the readers’ responsibility to be guessing what the authors want to communicate.

L 92, explain what are "imputed and actual values".  What is the difference between these two values?

L 97 use "were" instead of "are".  Overall, you should describe your methods in the past tense, because you are reporting something that you did in the past.

L 121, you should mention equations in sequence, that is, "Equations 1 and 2".

Section "2.2.2. Outlier Treatment", if I understood correctly, the outliers were removed from the dataset that was used for modeling?  Please be specific.

L 149, change "70%" for "Seventy percent".  It is not correct to start a sentence with digits.

I find it hard to understand how is it that the harvest was measured as kg per meter.  It seems that the correct thing to report would be kg per m^2.  Otherwise, it is not clear the dimension that relates to the mass of insects produced.

In Figure 4, the order of months in the x-axis is unclear. I was expecting to see a logical sequence of months, but the order that the months are presented in the figure has no logic.  It is necessary that the panels in Figure 4 are labeled (i.e., a, b and c), and each panel is described in the results and interpreted in the discussion.

I think that the text in lines 176-202 is hard to follow and understand because at this stage I cannot fully understand what exactly, the "Parameter Values" refer to.  At this stage, it would be worth reminding readers what, exactly, we are looking at in figure 5.  How does the "Parameter Values" can be interpreted?

L 206, you need to spell in full and describe in the methods section RMSE, MAE, and MAPE, and explain R-squared and Adjusted R-Squared, before you came up with these metrics in the results section.

Figure 6 is not a figure, is a Table.  Correct.

Even though there is nothing wrong with presenting results and discussion together, I strongly recommend authors to separate and improve the description and interpretation of these sections.  In my opinion, this will allow for a better understanding of these sections.

L 211-216, the discussion is around the top five important features identified by the random forest regression models (Figure 8).  But one result that stands out in Figure 8 and is overlooked by the authors, is the very large difference of relative importance values between Phase2 Laoded Length (m) and all other parameters.

I insist to separate the different pieces of information into their corresponding sections, as this revised manuscript still contains text in the results section that is related to the methods and thus should be placed in the methods section (e.g., lines 219-222).

It is not clear in the methods section how the comparison was made between the models fitted in this study and models from previous studies.  This is essential to understand the results reported in lines 219-228.

Figure 9 is not a figure; it is a Table.

L 235, "feature" or "future"?

The authors declare that no competing interests exist, but it is not stated who provided the funding for the research.  This needs to be included.

The manuscript lacks the “Data Availability Statement”, but I suggest authors to have their datasets in a format suitable for review.

Author Response

Reviewer 2 Report (New Reviewer)

This paper was framed very broadly in the context of utilizing BSF to solve an array of world problems. The application in which the work was explored was much more narrow and therefore I beleive the introduction is not appropriate/makes false claims about what is being invesitgated. I request that the authors reframe the paper in the context of production efficiencies and optimization through modeling processes. While much of what is in the introduction is true and has potential to contribute to the societal challenges related to population growth and hunger, this paper does not make that connection and looks at one small portion of the BSF contribution to this problem. It is false advertisement for the context of the study.

The other major concern I have is in regards to the feeding rate tha twas used for this facility and how that was incorporated into the work. There are plenty of studies that not only reinforce the importance of the nutritional quality of the feed (Phase 1 Formulation SW and Phase 2 Formulation B in this paper) which was found to be an important factor in predicting the yield, but the rate at which they are fed does not seem to be included ANYWHERE. Please clarify and address this with the models.

Broadly, the authors conclude that the random forest regressor performed the best and better than previous methods of predicting yield. I am not convinced that this is economically significant and supported by the data presented. More time in the discussion needs to be spent describing the specifics about the metrics used to make this determination and related to actual differences in yield and ecnomic potential int he production faciity itself. 

I would like to see these three concerns addressed in a revised version of this paper.

Round 2

Reviewer 1 Report (Previous Reviewer 1)

You can see the effort of the authors in revising their manuscript and the many changes they made.  I feel that many of my previous comments were satisfactorily addressed.  While others were not.  For example, the authors intend for us readers to assume that the way Figure 4 should be interpreted is by comparing each plot with one another.  But this is rather unusual.  If authors want to compare three variables, they should use analysis and graphical representation adequate for this.  On the other hand, in my previous review, I suggested to the authors that they separate the results and discussion sections, and in their response to reviewers’ letter the authors state that "we have improved and separated the appropriate contents of... the Results and Discussion Section", but this is not true because the results and discussion sections are still joined in a single section.  It seems to me that perhaps the authors have problems with the use of English, and found it difficult to express themselves adequately.  So, I will not assume a bad intention of the authors in answering that they took a certain action when they did not.  But I want to strongly caution the authors to be careful with this because it is unacceptable behavior to say they did something when they did not.

L 72, indicate how the quantity and weight of larvae were measured.  Include mention of apparatus/equipment used.

L 73, heading of Table 1 and elsewhere where you mention "variables", do you mean the input variables (i.e., the variables that were modeled)?  Please be specific.

L 78, Again, you need to indicate how the larvae were weighed.  Give detail about the balance/scale used.

There are several instances of an incorrect format of the manuscript.  Please stick to the instructions for authors indicated by the journal.  For instance, make effective mention of "section", "subsection" and "subsubsection", as indicated in the template.  Tables 4 and 5 are inserted as images, again, correct this following instructions in the template.  The simple summary is missing.  So many oversights on the part of the authors are not good because they suggest that they also prepared their manuscript with little care.

In Table 1: variable 8, just define what is "YL".  Variable 14 "Quantity Harvested", is described as kg, but kg is a measure of mass.  Quantity should be expressed as the number of larvae.  Or change the name of the variable to "Mass of larvae harvested", or another name.

L 237, as I bring up to the attention of the authors in my previous review, how is it that the quantity harvested is expressed by linear meter?  This has no sense to me and I think many readers will be confused.

L 237-239 "It is observed that for the rearing process, there is an increase in harvest as the bed length decreases" - Figure 4 does not show this.  Figure 4 shows how the variables "Monthly Mean Harvest (kg/m)", "Phase1 Feed Depth (in)", and "Phase1 Bed Length (m)", change as a function of time (months).

L 245, "It’s" is not formal scientific writing.  Correct here and elsewhere in the manuscript where it applies.

L 246, "this is also observed in Figure 4", this is misleading.  Again, Figure 4 does not show a correlation between variables.  Authors need to effectively present their results.

Author Response

Reviewer 2 Report (New Reviewer)

Thank you for making the suggested improvements to this manuscript. I believe it has improved the paper and clarified the implications of the findings. Please look at the grammar in the newly-written portions of the paper.

Round 3

Reviewer 1 Report (Previous Reviewer 1)

The authors addressed my previous comments satisfactorily.  Only a couple of minor comments:

L 90-91, 82-83, Give details on the weighing balance (brand, model, supplier, country).

Define target and input variables the first time they are mentioned.

I have no further comments for the authors, except to ask them to make a thorough and conscientious revision of the manuscript to avoid additional errors or details that have escaped the reviewers' view.  And present all the information in your manuscript accurately and clearly.

Author Response

This manuscript is a resubmission of an earlier submission. The following is a list of the peer review reports and author responses from that submission.

Round 1

Reviewer 1 Report

The manuscript "Application of Machine Learning Techniques to Discern Optimal Rearing Conditions for Improved Black Soldier Fly Farming", addresses a timely and relevant topic of research in the field of insect-rearing science and technology.  I believe this manuscript could potentially be of great interest to the readership of Insects but first, authors should perform a major revision of their paper to make their research more understandable.  In its current form, the overall structure of the manuscript is rather atypical and hard to follow.  In some instances, I was confused about whether this was a review or a research article.  The results and discussion sections are missing from the manuscript.  I strongly suggest authors to follow the conventional IMRAD (Introduction, Methods, Results, and Discussion) approach for scientific writing.   This will require a major restructuring of the manuscript.  I was taken aback that the editorial team did not spot this in their initial review of the manuscript since the instructions for authors and manuscript template indicate that research articles should follow this structure.

At this point, I was unable to make a fair assessment of the research results and of the interpretation authors gave to their results, as these are somehow disguised within the methods section.  Authors need to present their research in a logical and easy-to-follow structure.

Below, I list some specific comments that I believe can improve the quality of the manuscript.

L 2-4, Suggest rewriting as: "The use of insects, particularly the Black Soldier Fly (BSF) Hermetia illucens (L.) (Diptera: Stratiomydiae), as a source of feed stands out due to its sustainability and reliability".

L 4-7, Suggest rewriting as: "The Black Soldier Fly larvae (BSFL) has the ability to convert organic substrate to high-quality biomass rich in protein for animal feed, it can produce biodiesel and bioplastic, and has high biotechnological and medical potential".

L 7, Suggest "However, current BSF larvae production is low to meet the industry’s needs".

L 9, After the sentence ending with "BSF Farming.", add information about the input variables and response variables that were modeled.

L 9-11, For those of us who are not experts on machine learning, these lines have low significance.  Why should someone care about forest regressor statistics?  I suggest authors highlight the most relevant results from a biological perspective rather than a statistical perspective.

L 14, Explain "optimal ranges". 

In the introduction, there is an overall lack of citations to support the arguments presented.  Authors should cite appropriate references to support their claims.  For instance, in the first paragraph, of four sentences, only one of them has a citation.

L 25, Spell out BSF and include the scientific name of the species the first time you mention it in the main body of the article.

L 40-42, This sentence sounds like the aim of the work but the way it is presented is difficult to understand.  Revise English and clarity.

L 46, Define optimize/optimal in the context of the study.

L 47, Suggest "larval".

L 48, After "previous works" add "covering the topics of...".

L 48-49, What about the results?  In which section are they presented? 

L 50-197, This format of a literature review is not suitable for research articles.  Move the section "2. Literature Review" to a Supplementary material or an Appendix.

In Table 1, I am wondering if it would not be better to present the variables ordered by Phase (i.e., first the variables from Phase 1 followed by variables from Phase 2)?

L 245, Briefly describe what the flooring and capping approaches consist of so that readers can better understand the logic of your analyses.

L 249-251, What is the relevance of the analyses of the percentiles shown in Table 3.  Please explain.

L 258, what is "hot encoding".  Explain.

L 262, Explain "Feature splitting".

L 266, "Kgs" is an error.  The symbol of a kilogram is "kg".  Correct here and elsewhere in the manuscript.

L 273, If I understood correctly, from the section "3.3. Data Exploration" in the methods section, do you start to report results?  Please separate pieces of information into their corresponding section as indicated before following the IMRAD approach.

L 344, “generalized linear model"?

Reviewer 2 Report

The industrial uses of insects have become popular recently as the need for innovative solutions in managing food security programs.

The authors were done an innovative search, they emphasized interesting topics, so, I recommend accepting this article with minor modifications.

The correction, not limited to, 

1. The structure of the article should be re-formed in a sequence of Introduction, Materials and methods, Results, Discussion, and conclusion in addition to abstract, keywords, and references section

2.L75, the industrial waste should be specified 

3. L88 we can remove diet as the topic is talking about the optimal temperature 

4. in the humidity subsection, the authors should mention other compatible references 

5. In the pH subsection the mechanism of different pH should be mentioned, L120-122

6. All typo mistakes should be proper rewriting for example not limited to, L.133  BSF instead of BFS

7. Explanation of all experimental solutions should be mentioned, for example, L.136-140. 

8. the mathematical and computational models sub-section must be restructured as it is in the form of a paper summary. It should present in a comprehensive way

9. the references should be updated till the year 2023   

10. In the methodology section, the explanation of choosing the period of data collection should be mentioned and explained in the line highly scientific methodology